# VIBEFACE - Video and Image Biometric Dataset for Evaluation of Faces

## Abstract

Face verification has become a critical component of modern security systems, with facial biometrics widely adopted in sectors such as banking, mobile device authentication, and secure access to applications. To develop and evaluate robust biometric systems, a comprehensive dataset is required. While existing facial biometric datasets often address variations in pose, lighting, and demographics, they rarely capture realistic operational settings such as electronic Know Your Client (eKYC) procedures — a critical requirement for financial and regulatory compliance. To address this gap, we present VIBEFACE, a novel dataset specifically designed to support face verification in eKYC and related scenarios. VIBEFACE comprises 2,250 high-quality facial images (1,250 standardized and 1,000 selfie photographs) and 1,550 short videos, including both selfie recordings and sequences that explicitly mimic eKYC workflows. Data were collected using mobile device cameras from 50 diverse subjects, ensuring coverage and balance of demographic attributes, including gender, race, and age. By integrating realistic eKYC sequences with diverse visual conditions and strict adherence to ethical and legal standards, VIBEFACE establishes a new benchmark for evaluating the robustness and fairness of biometric verification systems in practical, compliance-driven environments.

## 1 Introduction

Facial verification is one of the most widely adopted biometric modalities for automatic identity authentication in real-world applications Rattani & Derakhshani (2018), including mobile device security Gimba & Ariffin (2024), banking systems Syed et al. (2024) Marco (2023), border control Hidayat et al. (2024). Especially, electronic Know Your Client (eKYC) procedures are gaining in popularity. Its growing popularity stems from its contactless and hygienic nature, usability, and increasing accuracy of AI-driven face recognition technologies.

Know Your Client has emerged as a cornerstone of secure digital transactions, enabling institutions to verify user identities without in-person interaction. Unlike traditional authentication settings, eKYC sessions often involve users recording short videos under unconstrained conditions — at home, in variable lighting, and across heterogeneous mobile devices. These videos capture natural facial behaviors such as blinking, head turning, and changes in expression, which are essential for both identity verification and liveness detection. Given their real-world relevance, datasets that include eKYC-style recordings can significantly benefit research by providing a benchmark for evaluating how verification algorithms generalize across demographics, devices, and interaction patterns.

Numerous studies have shown that biometric systems are vulnerable to performance degradation when exposed to demographic variations Terhörst et al. (2021) and non-ideal recording conditions. In particular, differences in race, age, and gender have been shown to impact verification accuracy significantly Sun et al. (2017), often leading to biased outcomes and reduced reliability for underrepresented groups. This highlights the importance of constructing datasets that encompass not only broad demographic diversity but also varied capture modalities.

Despite the proliferation of facial recognition datasets, publicly available resources remain limited in two critical aspects. First, most existing datasets focus on still images or controlled short video clips, which fail to capture the natural dynamics of eKYC interactions. Second, demographic imbalances and a lack of diverse acquisition settings hinder the ability of current benchmarks to support fair

and generalizable verification systems. To the best of our knowledge, there are no publicly available datasets that include authentic eKYC-style facial videos alongside still images.

To address these challenges, we introduce VIBEFACE, a novel multimodal dataset that integrates high-quality photographs with eKYC-style short videos from 50 participants, balanced across gender, race, and age groups. The dataset comprises 2,250 still images and 1,550 videos, captured under five distinct environmental conditions and across multiple consumer-grade smartphones. By emphasizing demographic fairness, cross-device variability, and real-world interaction dynamics, VIBEFACE provides a unique and much-needed resource for advancing research in robust, unbiased, and practically deployable facial verification systems. All data collection procedures adhered to ethical guidelines, with informed consent obtained from all participants. We aim to provide the research community with a comprehensive and demographically rich resource for benchmarking facial verification systems that are robust, fair, and generalizable across diverse populations.

## 2 RELATED WORK

Recent advances in face recognition have been largely driven by the combination of deep learning techniques and the availability of large-scale datasets containing facial images. Datasets such as VGGFace2 Cao et al. (2018), MS-Celeb-1M Guo et al. (2016), and WebFace260M Zhu et al. (2021) were constructed by automatically crawling publicly accessible images from the Internet. In most cases, these images were collected without the knowledge or explicit consent of the individuals depicted, raising serious concerns regarding legality, ethics, and privacy.

In response to increasing public, academic, and regulatory scrutiny, as well as growing awareness of the associated risks, several widely used datasets, including MS-Celeb-1M, VGGFace2, and MegaFace Kemelmacher-Shlizerman et al. (2016), have been officially withdrawn.

Moreover, many Internet-based datasets are poorly aligned with real-world authentication challenges. Today, facial biometrics are commonly used on non-specialized consumer devices, such as smartphones. In response, several dedicated datasets have been developed to better reflect practical conditions and realistic scenarios for the use of biometrics in authentication systems. For example, MOBIO McCool et al. (2012), Replay-Mobile Costa-Pazo et al. (2016), and OULU-NPU Boulkenafet et al. (2017) include variations in lighting conditions, while other databases, such as WMCA George et al. (2019) and HQ-WMCA Heusch et al. (2020), incorporate factors that can affect verification performance, such as the presence of eyeglasses. From an ethical and fairness perspective, demographic metadata inclusion is essential. The MobiBits Bartuzi et al. (2018) dataset provides information on participants' gender and age, enabling more comprehensive analysis. Beyond the availability of demographic data, ensuring its balance is equally important. The SOTERIA Ramoly et al. (2024) dataset achieves gender balance and provides substantial racial diversity. However, despite efforts to include a broad age range, middle-aged and older individuals remain underrepresented. The comparison of datasets is shown in Table 1.

The VIBEFACE dataset was created to address key limitations in existing resources. It provides videos depicting eKYC procedures while ensuring better demographic balance, especially across age, race, and gender. It includes greater variation in lighting and occlusions, such as eyeglasses. All data was ethically and legally sourced. The dataset also reflects realistic authentication scenarios, featuring still images and video recordings captured across multiple consumer devices.

## 3 DATASET

Data acquisition was conducted in a controlled studio environment, each session in a separate room specifically arranged to ensure consistent experimental conditions. Throughout the process, participants received standardized instructions and were continuously supervised by trained operators to maintain procedural uniformity and adherence to the defined recording protocols. Data were collected using widely available consumer-grade smartphones, including the Xiaomi Redmi Note 13, Apple iPhone 13, and Samsung Galaxy A35 5G. For smartphones running the Android operating system, images were originally captured in JPG format and videos in MP4 format. On iPhone devices, images were stored in HEIC format, while videos were recorded in MOV format. To ensure consistency across the dataset, all images were converted to JPG format and all videos to MP4 for-

Table 1: Comparison of facial biometric datasets. IDs - number of unique identities included, DD - demographic data, GB - gender balance, RB - race balance, AB - age balance.

| Dataset | IDs | Photos | Videos | eKYC | Glasses | DD | GB | RB | AB |
|---|---|---|---|---|---|---|---|---|---|
| MOBIO McCool et al. (2012) | 150 | | ✓ | | | | | | |
| Replay-Mobile Costa-Pazo et al. (2016) | 40 | ✓ | ✓ | | | | | | |
| OULU-NPU Boulkenafet et al. (2017) | 55 | | ✓ | | | | | | |
| MobiBits Bartuzi et al. (2018) | 53 | ✓ | | | ✓ | ✓ | | | |
| WMCA George et al. (2019) | 72 | | ✓ | | ✓ | | | | |
| HQ-WMCA Heusch et al. (2020) | 51 | | ✓ | | ✓ | | | | |
| Soteria Ramoly et al. (2024) | 70 | ✓ | ✓ | | | ✓ | ✓ | ✓ | |
| **VIBEFACE (Ours)** | 50 | ✓ | ✓ | ✓ | ✓ | ✓ | ✓ | ✓ | ✓ |

Figure 1: Demographic distribution of participants, featuring A: demographic pyramid and B: distribution of participants across four self-identified ethnic groups.

mat. Except for format conversion, no additional pre- or post-processing was applied to the data. All materials were captured in a vertical (portrait) orientation to reflect typical usage scenarios. All videos are provided in Full HD resolution (1920×1080 pixels), while images were captured at the maximum camera resolution, with a minimum size of 2316×3088 pixels.

## 3.1 DEMOGRAPHICS

A total of 50 individuals participated in the dataset acquisition process. Demographic and physical characteristics were collected through entry questionnaires, which included information on gender (male/female), age (numerical value), self-identified racial category (African, Caucasian, East Asian, South Asian), presence of facial hair (true/false), hair color (black, brown, blonde, red, gray, none), and facial piercings (excluding ear piercings; true or false). These attributes are provided in the metadata associated with each subject.

Participants wore their own clothing and were allowed to apply makeup at their discretion. The dataset includes 25 female and 25 male subjects, resulting in a 50:50 gender distribution. Participants' age ranges from 18 to 69. The distribution was designed to comply with the ISO Central Secretary (2011) standard. The gender and age structure is presented in Figure 1a.

Furthermore, VIBEFACE was balanced across four racial categories, with 13 African, 13 Caucasian, 12 East Asian, and 12 South Asian participants, respectively. We also ensured that the skin tones of participants reflect the whole spectrum of Fitzpatrick's scale. The racial composition is illustrated in Figure 1b.

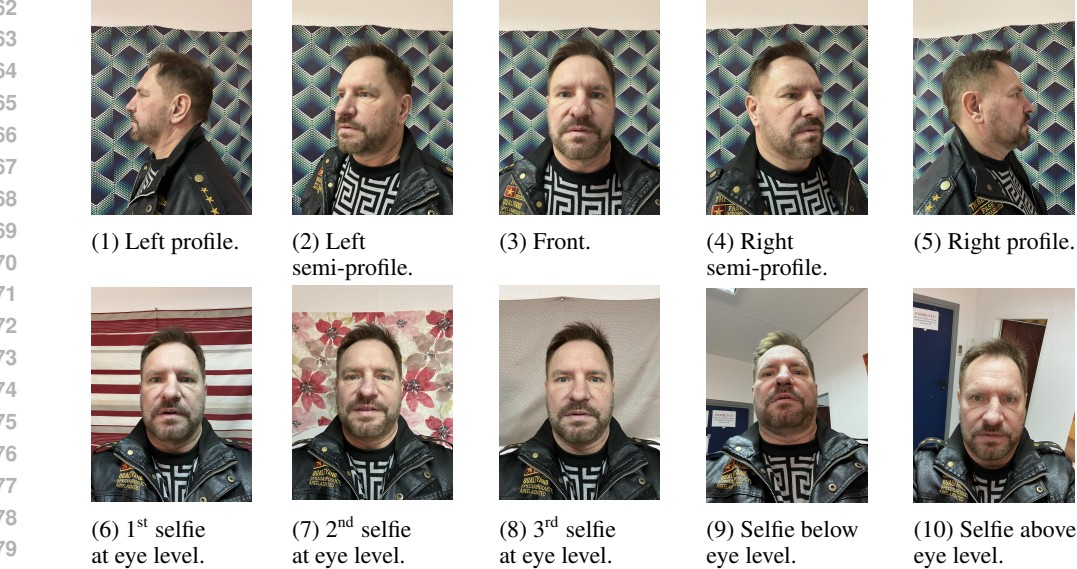

| (1) Left profile. | (2) Left semi-profile. | (3) Front. | (4) Right semi-profile. | (5) Right profile. |
| (6) 1st selfie at eye level. | (7) 2nd selfie at eye level. | (8) 3rd selfie at eye level. | (9) Selfie below eye level. | (10) Selfie above eye level. |

Figure 2: Standardized (1-5) and selfie photos (6-10) of the participant without eyeglasses in artificial light condition (session A). The numbers in parentheses refer to the scenario number.

## 3.2 SCENARIOS

We defined four scenario groups: standardized photos, selfie photos, selfie videos, and verification videos. In all scenarios, participants were instructed to maintain a neutral facial expression, position themselves directly in front of the camera aligned at eye level, and ensure that their face occupied approximately 30-60% of the frame width, unless explicitly specified otherwise by the scenario. Backgrounds were varied between successive scenarios to introduce additional diversity.

Standardized photos consist of five images taken by the operator using a rear camera: left profile, left semi-profile, front, right semi-profile, and right profile. Each scenario is assigned an identification number ranging from 1 to 5, respectively.

Each selfie photo scenario includes five images captured by the participant using the front-facing camera of a smartphone. The first three were taken at eye level (scenarios 6-8), while the remaining two were captured from lower and higher angles, respectively (scenarios 9 and 10), to introduce variability in facial perspective. Examples of standardized and selfie photos (scenarios 1-10) are shown in Figure 2.

The selfie video scenario involves one short recording that simulates a typical user behavior, where the participant activates the front-facing camera positioned below face level, gradually raises it to eye level, and holds this position for at least three seconds (scenario 11).

Verification videos are short recordings in which participants perform a sequence of actions commonly used in eKYC facial video verification tasks. These actions include:

- circular head rotation in both directions (scenario 12),
- tilting the head up and down and turning it left and right (scenario 13),
- blinking at least three times (scenario 14),
- changing facial expression (smiling and returning to a neutral expression) (scenario 15),
- opening the mouth wide and then closing it (scenario 16),
- partially covering the face with a hand (both vertically and horizontally) (scenario 17),
- sequentially touching various parts of the face with a finger: nose, chin, forehead, and cheek, returning to the initial position after each touch (scenario 18).

All verification videos were recorded using the smartphone's front camera operated by the participant. Frames from example eKYC videos are shown in Figure 3.

## 3.3 SESSIONS

To reflect real-world variability in facial appearance, we designed five acquisition sessions that differ in both lighting conditions and the presence or absence of eyeglasses. Lighting conditions are known to influence facial feature representation significantly and can differentially affect performance across ethnic groups, potentially introducing demographic bias Pangelinan et al. (2025). To account for this, we implemented four distinct lighting scenarios:

- **Artificial light:** provided by non-specialized indoor lighting sources. To simulate a range of common indoor environments, we alternated between cool white LED, warm yellow tungsten, and diffuse softbox illumination.
- **Flash:** generated by the smartphone's built-in flash.
- **Natural daylight:** sunlight entering directly through an uncovered window. Due to the acquisition being performed across multiple days and at different times, natural light intensity varied as a function of both diurnal and weather-related conditions.
- **Weak natural light:** diffuse daylight filtered through a partially covered window, with the subject positioned at least five meters from the light source.

The flash lighting scenario required the use of a back-facing camera. As a result, this session includes only standardized photographs performed by the operator.

The presence of eyeglasses, due to reflections and partial occlusion of facial features, can negatively impact facial verification performance. To investigate this factor, we conducted an additional session under artificial lighting conditions in which all participants wore glasses. During the dedicated 'glasses' session, participants who normally wear prescription glasses were permitted to use their own, while others were provided with non-corrective (zero-lens) eyeglasses to ensure consistency in occlusion conditions.

Moreover, when appropriate, participants were instructed to don or remove outerwear to mitigate the potential impact of clothing features on facial verification performance. The acquisition device was randomly chosen before each session. An overview of all sessions, along with the scenarios included in each, is presented in Table 2.

Table 2: Scenarios implemented across sessions. A "✓" sign indicates that a given scenario was included in the session.

| Session | Light conditions | Glasses | Scenarios | | | |
| --- | --- | --- | --- | --- | --- | --- |
| | | | 1-5 | 6-10 | 11 | 12-18 |
| A | Artificial light | No | ✓ | ✓ | ✓ | ✓ |
| B | Flash | No | ✓ | | | |
| C | Artificial light | Yes | ✓ | ✓ | | ✓ |
| D | Natural light | No | ✓ | ✓ | ✓ | ✓ |
| E | Weak natural light | No | ✓ | ✓ | ✓ | ✓ |
| | **Camera** | | Back | Front | Front | Front |

Across all sessions and scenarios, each participant contributed a total of 45 still images and 31 video recordings, resulting in an aggregate of 2,250 images and 1,550 videos in the dataset.

## 3.4 ETHICAL CONSIDERATIONS AND PRIVACY

The VIBEFACE dataset was fully collected in compliance with ethical research standards. Prior to data acquisition, each participant received detailed information regarding the purpose, scope, and potential dataset uses. Informed consent was obtained from all individuals who voluntarily agreed to contribute their facial images and videos for research purposes, including applications in biometric systems and artificial intelligence development. The agreement stands in compliance with

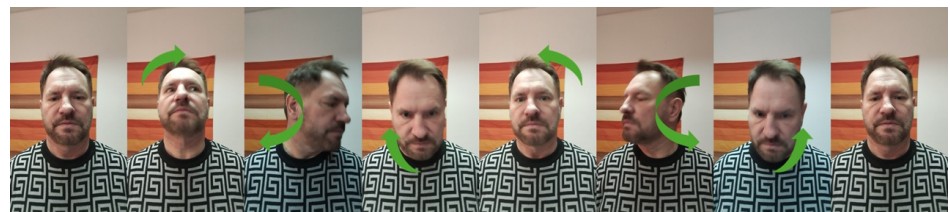

(12) Circular head rotation in both directions.

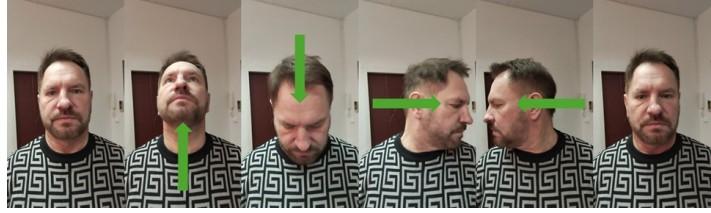

(13) Tilting the head up and down and turning it left and right.

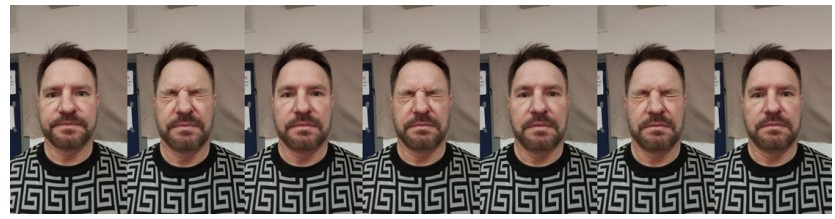

(14) Blinking at least three times.

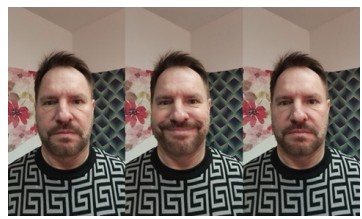

(15) Changing facial expression.

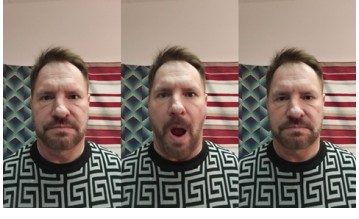

(16) Opening the mouth wide and then closing it.

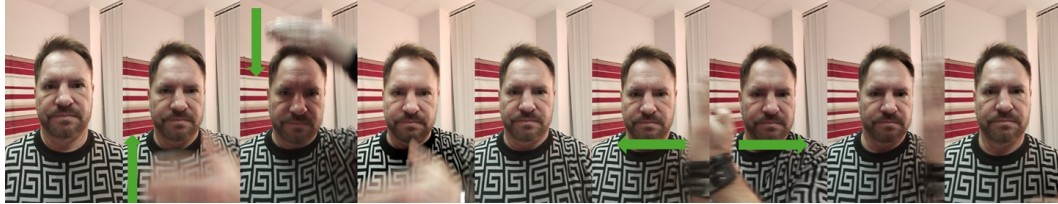

(17) Partially covering the face with a hand (both vertically and horizontally).

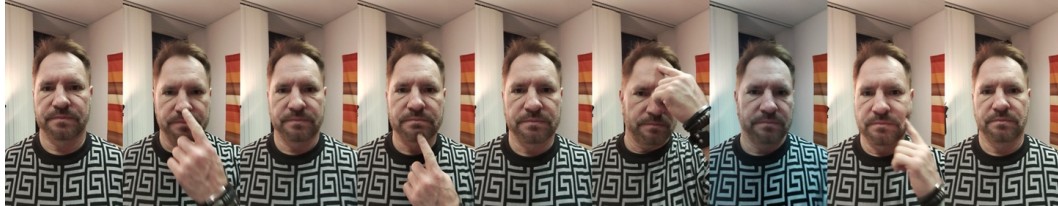

(18) Sequentially touching various parts of the face with a finger: nose, chin, forehead, and cheeks, returning to the initial position after each touch.

Figure 3: Frames retrieved from eKYC video scenarios performed by participant without eyeglasses in artificial light condition (session A). The numbers in parentheses refer to the scenario number. Green arrows were added to depict movement.

the European Union General Data Protection Regulation (GDPR) European Parliament and Council (2016) and AI Act European Parliament and Council (2024).

Participants were explicitly informed that their data may be made available to the research community under a controlled-access agreement for non-commercial, academic use only. All subjects had the right to withdraw their participation at any stage of the process, without any consequences. Only adult participants (18 years and older) were recruited, and efforts were made to ensure demographic diversity in terms of gender, race, and age to foster fairness in algorithmic benchmarking. No personally identifiable information, such as names and contact details, was stored or associated with the dataset to minimize the risk of misuse or unauthorized re-identification. All image and video files are anonymized and stored using randomized identifiers.

Data acquisition was supervised by trained operators in controlled settings to ensure participant safety and psychological comfort throughout the process.

## 3.5 LICENSING AND ACCESS

The VIBEFACE dataset is publicly available for non-commercial, research purposes only and can be accessed upon signing the Research Data License Agreement. The agreement form and additional information can be obtained from the official project website: `Paperunderdouble-blindreview`[1]. The dataset is released under a controlled-access license to ensure ethical and legal use of sensitive biometric data. Commercial applications, redistribution, and any attempt at re-identification are strictly prohibited.

## 4 BENCHMARK TASKS

To assess the usability of the dataset, we conducted two evaluation tasks: face detection and biometric face verification.

### 4.1 FACE DETECTION

Face detection plays a crucial role in numerous computer vision applications, particularly in biometric verification systems. Accurate localization of the facial region within the input image is a prerequisite for subsequent operations, such as feature extraction and comparison with a reference biometric template. The overall effectiveness of the identity verification process is therefore highly dependent on the precision of the detection module. This requirement becomes especially important in mobile or remote authentication systems, which must operate autonomously under unpredictable environmental conditions. Such systems are often exposed to challenges, including varying illumination, occlusions, diverse head poses, facial expressions, and fluctuating image quality.

To evaluate the impact of dataset diversity on face detection performance, we conducted a comparative analysis using three widely adopted face detection algorithms: MTCNN Zhang et al. (2016), RetinaFace Deng et al. (2020), and MediaPipe Bazarevsky et al. (2019). The evaluation was conducted using data from scenarios 1–10 and 12–16 across sessions A, C, D, and E. Scenario 11 was excluded due to incomplete coverage across sessions, while scenarios 17 and 18 were omitted because they involve occlusions that significantly reduce facial visibility. For video-based scenarios, frames were extracted at a sampling rate of 6 frames per second. Image scenarios were grouped into two categories: the first included scenarios 1, 2, 4, 5, 9, and 10, representing challenging poses; the second comprised scenarios 3 and 6–8, which reflect more favorable and commonly encountered conditions for face verification. Detection performance was quantified as the percentage of frames in which a face was successfully detected. Additionally, a demographic analysis was performed to assess detection robustness across sessions, gender, age groups, and racial categories.

Detection performance was quantified as the percentage of frames or images in which a face was successfully detected by the algorithm. The results are shown in Table 3.

---

[1]For reviewing purposes only, the dataset can be accessed via the following link: `https://tinyurl.com/3pwvx7tt` - temporary password: ahudQMtpATP-bKSiIt5jclNszbSuRbKGkLqOe8ME74YfEEN1TVQv9CflbtHxM75Lx

RetinaFace and MediaPipe consistently achieved the highest face detection rates across all scenarios, sessions, and demographic groups. While MTCNN remained effective overall, it exhibited greater sensitivity to variations in lighting and image acquisition conditions, with notably lower detection rates observed in session C.

Demographic analysis revealed minimal variation in detection rates across gender and age groups for MediaPipe and RetinaFace. However, MTCNN showed reduced detection performance among both younger and older participants, as well as in certain racial categories, particularly among individuals of African descent, while achieving its highest accuracy for East Asian subjects.

Dynamic scenarios (12–16) and image-based scenarios with non-frontal or off-angle poses (1, 2, 4, 5, 9, 10) introduced greater variability in head position and motion, moderately degrading detection for some models. The most significant drops occurred in scenarios involving circular head rotation (12), directional tilts (13), and challenging poses (1, 2, 4, 5, 9, 10), especially in sessions C and E. These sessions combined off-angle views with eyeglasses or weak natural light, resulting in substantial accuracy loss—particularly for MTCNN.

Table 3: Face detection results broken down by scenario, gender, age group, and racial category. Abbreviations: Scn. - Scenario, Afr. – African, Cauc. – Caucasian, EA – East Asian, SA – South Asian, OAV - Off-Angle Views (scenarios 1, 2, 4, 5, 9, 10), FV - Frontal Views (scenarios 3, 6-8).

| Scn. | Session | | | | Gender | | Age group | | | Race category | | | |
|------|-------|-------|-------|-------|-------|-------|-------|-------|-------|-------|-------|-------|-------|
| | A | C | D | E | ♀ | ♂ | 18-30 | 31-50 | 51-70 | Afr. | Cauc. | EA | SA |
| **MTCNN** | | | | | | | | | | | | | |
| OAV | 0.764 | 0.631 | 0.655 | 0.577 | 0.678 | 0.635 | 0.656 | 0.697 | 0.610 | 0.675 | 0.629 | 0.669 | 0.655 |
| FV | 0.970 | 0.882 | 0.935 | 0.856 | 0.918 | 0.904 | 0.878 | 0.964 | 0.890 | 0.812 | 0.911 | 0.984 | 0.943 |
| 12 | 0.861 | 0.728 | 0.785 | 0.755 | 0.829 | 0.732 | 0.751 | 0.851 | 0.745 | 0.705 | 0.784 | 0.877 | 0.763 |
| 13 | 0.826 | 0.663 | 0.754 | 0.699 | 0.778 | 0.694 | 0.710 | 0.787 | 0.711 | 0.645 | 0.738 | 0.813 | 0.752 |
| 14 | 0.954 | 0.885 | 0.944 | 0.934 | 0.949 | 0.908 | 0.913 | 0.965 | 0.906 | 0.842 | 0.913 | 0.990 | 0.979 |
| 15 | 0.955 | 0.823 | 0.904 | 0.912 | 0.928 | 0.870 | 0.872 | 0.956 | 0.867 | 0.783 | 0.893 | 0.989 | 0.941 |
| 16 | 0.967 | 0.877 | 0.946 | 0.897 | 0.940 | 0.904 | 0.904 | 0.959 | 0.902 | 0.805 | 0.928 | 0.994 | 0.970 |
| **RetinaFace** | | | | | | | | | | | | | |
| OAV | 1.000 | 1.000 | 1.000 | 1.000 | 1.000 | 1.000 | 1.000 | 1.000 | 1.000 | 1.000 | 1.000 | 1.000 | 1.000 |
| FV | 1.000 | 1.000 | 1.000 | 1.000 | 1.000 | 1.000 | 1.000 | 1.000 | 1.000 | 1.000 | 1.000 | 1.000 | 1.000 |
| 12 | 0.999 | 0.990 | 0.991 | 0.977 | 0.989 | 0.989 | 0.985 | 0.999 | 0.984 | 0.994 | 0.996 | 0.981 | 0.984 |
| 13 | 0.997 | 0.986 | 0.999 | 0.990 | 0.999 | 0.988 | 0.985 | 0.999 | 0.998 | 0.997 | 0.986 | 0.998 | 0.992 |
| 14 | 1.000 | 1.000 | 1.000 | 1.000 | 1.000 | 1.000 | 1.000 | 1.000 | 1.000 | 1.000 | 1.000 | 1.000 | 1.000 |
| 15 | 1.000 | 0.999 | 1.000 | 1.000 | 1.000 | 0.999 | 0.999 | 1.000 | 1.000 | 1.000 | 0.999 | 1.000 | 1.000 |
| 16 | 1.000 | 1.000 | 1.000 | 1.000 | 1.000 | 1.000 | 1.000 | 1.000 | 1.000 | 1.000 | 1.000 | 1.000 | 1.000 |
| **MediaPipe** | | | | | | | | | | | | | |
| OAV | 0.975 | 0.924 | 0.957 | 0.968 | 0.959 | 0.953 | 0.961 | 0.956 | 0.948 | 0.969 | 0.960 | 0.948 | 0.945 |
| FV | 0.995 | 0.992 | 1.000 | 1.000 | 0.998 | 0.996 | 1.000 | 1.000 | 0.988 | 1.000 | 0.987 | 1.000 | 1.000 |
| 12 | 0.976 | 0.947 | 0.963 | 0.952 | 0.973 | 0.945 | 0.931 | 0.980 | 0.970 | 0.975 | 0.947 | 0.980 | 0.938 |
| 13 | 0.964 | 0.955 | 0.980 | 0.967 | 0.987 | 0.946 | 0.947 | 0.982 | 0.975 | 0.976 | 0.942 | 0.990 | 0.963 |
| 14 | 0.980 | 0.984 | 0.981 | 0.990 | 1.000 | 0.966 | 0.955 | 1.000 | 1.000 | 1.000 | 0.940 | 1.000 | 1.000 |
| 15 | 0.986 | 0.981 | 0.987 | 1.000 | 1.000 | 0.977 | 0.970 | 1.000 | 0.999 | 0.999 | 0.957 | 1.000 | 1.000 |
| 16 | 0.979 | 0.982 | 0.987 | 0.990 | 0.999 | 0.970 | 0.961 | 0.999 | 0.998 | 0.999 | 0.945 | 0.999 | 0.999 |

## 4.2 FACE VERIFICATION

In practical applications, particularly on mobile and remote platforms, face verification systems must demonstrate both high accuracy and robustness to a wide range of environmental and demographic variations. To support such evaluation, our dataset includes a wide range of challenging conditions and scenarios. We evaluated two state-of-the-art face verification models: ArcFace Deng et al. (2019) and MagFace Meng et al. (2021). For testing, a frontal image from the flash session (Scenario 3, Session B) was used as the reference sample, emulating a typical document-based authentication setup. As query samples, we used the same images and videos employed in the face detection experiment. Verification was considered successful when the similarity score exceeded a

fixed threshold of 0.5. Performance was measured as the percentage of frames in which the face was correctly authenticated. Results are presented in Table 4.

Our results demonstrate that ArcFace consistently outperformed MagFace across scenarios, sessions, and demographic groups. MagFace showed greater sensitivity to environmental variations and consistently underperformed in both static and dynamic settings.

Demographic analysis revealed modest performance disparities. Both models performed slightly worse on the Caucasian subgroup. Additionally, female participants consistently achieved slightly higher verification rates than males. Interestingly, the youngest age group (18–30) yielded the lowest performance among age groups.

Consistent with the face detection results, the most challenging conditions for verification were observed in sessions C and E, corresponding to the presence of eyeglasses and weak natural light, respectively. These environmental factors negatively impacted both models, particularly MagFace, highlighting the need for robust solutions in unconstrained settings.

Table 4: Face verification results broken down by scenario, gender, age group, and racial category. Abbreviations: Scn. - Scenario, Afr. – African, Cauc. – Caucasian, EA – East Asian, SA – South Asian, OAV - Off-Angle Views (scenarios 1, 2, 4, 5, 9, 10), FV - Frontal Views (scenarios 3, 6-8).

| Scn. | Session | | | | Gender | | Age group | | | Race category | | | |
|------|-------|-------|-------|-------|-------|-------|-------|-------|-------|-------|-------|-------|-------|
| | A | C | D | E | ♀ | ♂ | 18-30 | 31-50 | 51-70 | Afr. | Cauc. | EA | SA |
| **ArcFace** | | | | | | | | | | | | | |
| OAV | 0.509 | 0.433 | 0.505 | 0.481 | 0.476 | 0.487 | 0.466 | 0.469 | 0.519 | 0.490 | 0.468 | 0.460 | 0.509 |
| FV | 1.000 | 0.995 | 1.000 | 1.000 | 1.000 | 0.998 | 1.000 | 0.996 | 1.000 | 0.995 | 1.000 | 1.000 | 1.000 |
| 12 | 0.809 | 0.604 | 0.730 | 0.652 | 0.731 | 0.665 | 0.656 | 0.724 | 0.721 | 0.712 | 0.678 | 0.742 | 0.669 |
| 13 | 0.642 | 0.498 | 0.628 | 0.572 | 0.606 | 0.564 | 0.567 | 0.580 | 0.614 | 0.588 | 0.555 | 0.591 | 0.614 |
| 14 | 0.980 | 0.974 | 0.972 | 0.979 | 0.997 | 0.955 | 0.947 | 0.989 | 0.998 | 0.994 | 0.927 | 0.990 | 1.000 |
| 15 | 0.981 | 0.957 | 0.979 | 0.978 | 0.991 | 0.956 | 0.944 | 0.996 | 0.987 | 0.991 | 0.922 | 0.997 | 0.988 |
| 16 | 0.941 | 0.879 | 0.950 | 0.956 | 0.989 | 0.874 | 0.905 | 0.943 | 0.952 | 0.936 | 0.897 | 0.965 | 0.934 |
| **MagFace** | | | | | | | | | | | | | |
| OAV | 0.274 | 0.262 | 0.298 | 0.296 | 0.305 | 0.260 | 0.273 | 0.286 | 0.290 | 0.267 | 0.272 | 0.285 | 0.308 |
| FV | 1.000 | 1.000 | 0.995 | 1.000 | 1.000 | 0.998 | 0.997 | 1.000 | 1.000 | 0.995 | 1.000 | 1.000 | 1.000 |
| 12 | 0.679 | 0.526 | 0.610 | 0.548 | 0.612 | 0.569 | 0.549 | 0.637 | 0.590 | 0.562 | 0.539 | 0.678 | 0.594 |
| 13 | 0.530 | 0.445 | 0.527 | 0.492 | 0.526 | 0.472 | 0.479 | 0.511 | 0.509 | 0.491 | 0.460 | 0.523 | 0.530 |
| 14 | 0.980 | 0.963 | 0.981 | 0.981 | 1.000 | 0.951 | 0.947 | 0.990 | 0.996 | 0.986 | 0.925 | 1.000 | 1.000 |
| 15 | 0.981 | 0.926 | 0.971 | 0.970 | 0.992 | 0.932 | 0.928 | 0.991 | 0.973 | 0.953 | 0.902 | 1.000 | 0.999 |
| 16 | 0.849 | 0.757 | 0.878 | 0.876 | 0.923 | 0.757 | 0.815 | 0.860 | 0.850 | 0.862 | 0.796 | 0.854 | 0.854 |

## 5 CONCLUSIONS

In this paper, we introduced VIBEFACE, a novel dataset designed to support comprehensive evaluation and analysis of facial biometric algorithms under realistic conditions. To the best of our knowledge, it is the first publicly available database to include diverse video-based eKYC verification scenarios. We demonstrated the dataset's utility across two benchmark tasks: face detection and facial verification. Our findings highlight that many existing datasets lack demographic balance, which can result in biased model performance. VIBEFACE addresses these limitations by offering a legally compliant, demographically diverse, and application-relevant resource for advancing fair and robust face biometrics research. The potential applications of our dataset extend beyond the two evaluation tasks presented in this study.

Beyond the scope of the experiments presented here, VIBEFACE holds potential for broader applications. Its inclusion of diverse acquisition conditions and high-quality bona fide samples makes it well-suited for advancing research in presentation attack detection (PAD), as well as in emerging areas such as detecting injection attacks involving deepfakes.

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
