# OpenReview forum: "VIBEFACE - Video and Image Biometric Dataset for Evaluation of Faces"
_ICLR.cc/2026/Conference — ICLR 2026 Conference Withdrawn Submission_

### Official Review · Reviewer_NHTn · 2025-10-25

**Soundness:** 1
**Presentation:** 2
**Contribution:** 1
**Rating:** 0
**Confidence:** 4

**Summary:**

The paper proposes a dataset for the evaluation of eKYC applications. The authors proposed a small-scale facial biometric dataset containing ~2K images and ~1.5K videos from 50 subjects. The authors contribution is the specific focus on electronic Know Your Client (eKYC) scenarios, including short videos that mimic eKYC liveness-check workflows. The authors claim the dataset is demographically balanced across gender, race, and age and was collected ethically to address the legal and privacy concerns of large web-scraped datasets.

**Strengths:**

* The specific focus on eKYC applications is a notable and relevant direction, as this represents a critical real-world scenario for modern security systems that may not be well-represented in other public datasets.

**Weaknesses:**

The paper suffers from significant weaknesses in its scale, experimental design, and justification, which make its contribution as a new benchmark unconvincing.
### Limited Scale and Novelty,

 The primary weakness is the dataset's scale. With only 50 subjects, it is far too small for modern deep learning research and pales in comparison to other datasets (e.g., NerSemble v1 or v2) that can be utilized for the same applications. The authors' justification for not using existing large-scale datasets is also weak. Although it is correct that datasets like WebFace do not have some attribute labels, this is a solvable problem; we can simply label them using off-the-shelf classifiers or generate segmentation masks (e.g., to detect occlusions). This would yield a far more useful and scalable resource.

### Poorly Justified Benchmarks

The provided benchmarks are incomplete and lack rigor.

### Missing Attack Benchmarks,
The prominent factor for any modern verification benchmark is robustness to attacks. Why are there no benchmarks in this regard? The authors mention PAD and deepfakes as a potential application  but provide no evaluation. They must have a comparison with benchmarks like FaceForensics or Celeb-DF to be taken seriously.

### Missing Dataset Comparisons
There is no study of other datasets with a similar role under similar settings. Authors could have used a dataset like NerSemble, which also has demographic metadata, as a comparison point to concretely demonstrate VIBEFACE's advantages.


### Ungrounded Scenarios
 Where did the authors "brought" the eKYC actions (scenarios 12-18) from?  Are these actions common in eKYC applications or just an arbitrary list? The paper provides no citation or evidence that these are standard liveness-check procedures.
The paper is generally not detailed enough and there exists many ungrounded decisions.

### Lack of Experimental Detail

The paper lacks critical information. For example, in the verification benchmark, the authors state they used "ArcFace." ArcFace is a margin loss, not a model. What backbone architecture was used (e.g., vit)? What dataset was this model trained on? Without this, the results are uninterpretable and irreproducible.

### Questionable Benchmark Utility

 Is it a good benchmark when most of the numbers in the tables, using 6-year-old models, are all quite high (e.g., 1.000 for 'FV' although we acknowledge the drop for `OVA`)?  This suggests the dataset is not challenging. It is important to know what type of model authors have used. Would the results still be so high when using a strong backbone trained on a large dataset like WebFace12M?
### Arbitrary Threshold,

 There are so many ungrounded decisions. For instance, the authors state they "put the thershold to 0.5" for verification. Why this value? This arbitrary choice is not standard practice and renders the accuracy percentages reported in Table 4 meaningless. The authors should have chosen various thresholds.

**Questions:**

See the weaknesses

---

### Official Review · Reviewer_sshd · 2025-10-26

**Soundness:** 2
**Presentation:** 2
**Contribution:** 1
**Rating:** 0
**Confidence:** 5

**Summary:**

The paper presents a novel dataset, namely VIBEFACE, by capturing the facial images and videos reflecting eKYC scenarios needed in the current time for user verification. The dataset has been collected from 50 subjects covering diverse demographic entities. The dataset has been benchmarked with several existing algorithms covering face detection and the verification task of face recognition.

**Strengths:**

The primary contribution of the paper is the creation of a novel dataset reflecting eKYC settings.

The paper is easy to read and follow, and the contribution might be helpful in digital security.

**Weaknesses:**

- One of the primary weaknesses of the paper is the limited dataset sizes, not only in terms of the number of subjects but also in terms of the limited acquisition devices and the limited representation of the demographic entities.
- Second, there is no technical novelty in the paper. Moreover, the observations presented in the paper are well known, that the majority of the existing face verification algorithms suffer under pose and expression settings. Further, pose, expression, and illumination robust architectures are not explored. In other words, only 2 verification architectures are employed.
- As demonstrated by the results in Table 3, face detection is not a serious challenge in the eKYC scenarios, especially when RetinaFace and MediaPipe have been used. The lowest value reported is 0.977 for RetinaFace and 0.924 for MediaPipe.
- It is not clear how the verification has been performed. Is it per video or per frame? If it is per video, then how the recognition has been performed and if it is per frame, then how many frames are correctly recognised, and do we have to recognise all the frames for a true verification?
- It is mentioned that sessions C and E are the most challenging settings, which is not true. If it is the case, then it must be with scenario FV, 14-16.
- Scenario 12-13 might be the most challenging, but there is no explainability or interpretability of this observation.
- What about the scenarios 17-18 showcased in Figure 3, but neither Table 3 nor Table 4 has results related to them.

**Questions:**

It would be great if the responses to the following questions, along with the weaknesses raised above, could be provided:

- The rationale behind the selection of benchmarking architecture, especially the verification network and threshold, is not clear. They are not only old but also does not robust to the scenarions listed in the proposed dataset.

- As demonstrated by the results in Table 3, face detection is not a serious challenge in the eKYC scenarios, especially when RetinaFace and MediaPipe have been used. If it is true, then should we really worry about eKYC face verification issues?

- It is not clear how the verification has been performed. Is it per video or per frame?
- It is mentioned that sessions C and E are the most challenging settings, which is not true. If it is the case, then it must be with scenario FV, 14-16. Any comments on this point?
- The paper lacks explainability or interpretability of the results reported. Without the mentioned properties, the paper is merely another paper with limited size and known observations.
- What about the scenarios 17-18 showcased in Figure 3, but neither Table 3 nor Table 4 has results related to them.

---

### Official Review · Reviewer_qq4c · 2025-10-31

**Soundness:** 1
**Presentation:** 2
**Contribution:** 1
**Rating:** 0
**Confidence:** 5

**Summary:**

The authors propose an evaluation benchmark for face recognition using mobile devices.
They collected a few thousand images and videos from a handful of subjects in several controlled scenarios using a few different devices.
Using three face detection and two face recognition models, the authors indicated differences in performance between different genders, ethnicities and age groups.

**Strengths:**

The dataset collection followed GDPR and AI Act standards.

**Weaknesses:**

1. The motivation for the collection of the dataset does not seem to follow its intended use:

   a) In line 39, the authors require that eKYC data involves "short videos under unconstrained conditions". The reviewer was surprised to read in line 100 and section 3.2 that data was acquired in a controlled studio environment, using controlled and scripted capturing scenarios, which defeats its purpose.

   b) While the authors claim to have collected an age-balanced dataset, Figure 1A reveals that this is not the case. The age ranges differ between the three groups: (12, 20, 20). This is a common trick as described in the book "Lying with Statistics". Such wrongful statistic tricks necessitate the rejection of the paper.

   c) The distribution into the four groups (Caucasian, African, East Asian, South Asian) seems arbitrary and does not follow other definitions of ethnicity groups.

   d) While the authors wrongfully claim that "to the best of our knowledge, there are no publicly available datasets that include authentic eKYC-style facial videos alongside still images", in their related work they cite exactly such datasets, including the publicly available MOBIO and its extension [Khoury2014], while missing others like MOBBio [Sequeira2014]. At least the former contains gender annotations.

   e) In the conclusion, the authors state that "many existing [evaluation] datasets lack demographic balance, which can result in biased model performance", but the bias in the model is not caused by imbalanced evaluation datasets. Also, the authors claim that the dataset would be useful for presentation attack detection, but there is no single presentation attack included.



2. The evaluation of different face detectors and face recognition algorithms is clearly wrong:

   a) Considering the quality of the facial images, as shown in figures 2 and 3, any face detector as evaluated by the authors would clearly be able to detect most -- if not all -- of the faces. The authors likely have used them wrongly, especially MTCNN, for example, by selecting too strict detection thresholds (which are not specified by the authors).

   b) Reporting detection rate as "the percentage of frames in which a face was successfully detected" (line 370, and repeated in line 373) is not appropriate and does not follow its use case. For eKYC applications, it is sufficient that one or a few faces **per sequence** are detected. Hence, detection performance is likely to be 100% for all cases.

   c) Similarly, recognition accuracy should consider the correct identification of the whole sequence, not on an image level. Typical ways include to combine features extracted from different frames of the sequence [Khoury2014], modern systems just average embeddings across all detected faces.

   d) Using a fixed threshold of 0.5 (line 432) for verification is clearly and absolutely wrong. Thresholds need to be estimated (on a large-scale dataset that includes similar conditions as the test dataset, or on the test dataset itself), by assessing False Match Rates of $10^{-3}$ or lower. Thresholds also differ between image-based and template-based evaluations, as well as between face recognition models.

   e) Given the restrictions above, there is no need to exclude different scenarios (11, 17, 18; see section 4.1) from the detection and recognition evaluation.

   f) Demographic evaluation needs to follow standard evaluation metrics as defined in the ISO standard ISO/IEC 19795-10. The size of the proposed dataset is far too small to provide a reliable evaluation of bias.


3. There are some minor issues:

   a) There is only one human race. The authors should make use of proper terms, such as "ethnicity" or "skin tone".

   b) The citation style should be adapter throughout the paper, making proper use of \cite, \citep and \citet.

[Sequeira2014] Sequeira, Monteiro, Rebelo, Oliveira: "MobBIO: A multimodal database captured with a portable handheld device," VISAPP 2014

[Khoury2014] Khoury, El Shafey, McCool, Gunther, Marcel: "Bi-modal biometric authentication on mobile phones in challenging conditions", IVC 2014

[Pereira2022] de Freitas Pereira, Schmidli, Linghu, Zhang, Marcel, Gunther: "Eight Years of Face Recognition Research: Reproducibility, Achievements and Open Issues" arXiv, 2022

**Questions:**

Why would we need another small-scale evaluation benchmark? The number of samples in the proposed dataset are far too small to provide reliable performance evaluation for face detection and recognition methods, not even talking about biometric fairness evaluations.

Since the evaluation of the detection and recognition methods is flawed, the utility of the proposed dataset is not verified. Related research [Pereira2022] on other mobile datasets (MOBIO) have shown that verification is solved for such high-quality benchmarks by modern face recognition methods.

---

### Note · Authors · 2025-12-01

I have read and agree with the venue's withdrawal policy on behalf of myself and my co-authors.